# Design and Manufacturing of Equipment for Investigation of Low Frequency Bioimpedance

**DOI:** 10.3390/mi13111858

**Published:** 2022-10-29

**Authors:** Lucian Pîslaru-Dănescu, George-Claudiu Zărnescu, Gabriela Telipan, Victor Stoica

**Affiliations:** Laboratory of Sensors/Actuators and Energy Harvesting, National Institute for Research and Development in Electrical Engineering ICPE-CA, 030138 Bucharest, Romania

**Keywords:** low frequency bioimpedance, bioelectrical resistivity, electronic conditioning system, current injection electrode, electrode of electric potential

## Abstract

The purpose of this study was to highlight a method of making equipment for the investigation of low frequency bioimpedance. A constant current with an average value of *I* = 100 µA is injected into the human body via means of current injection electrodes, and the biological signal is taken from the electrodes of electric potential charged with the biopotentials generated by the human body. The resulting voltage, Δ*U* is processed by the electronic conditioning system. The mathematical model of the four-electrode system in contact with the skin, and considering a target organ, was simplified to a single equivalent impedance. The capacitive filter low passes down from the differential input of the first instrumentation amplifier together with the isolated capacitive barrier integrated in the precision isolated secondary amplifier and maintains the biological signal taken from the electrodes charged with the undistorted biopotentials generated by the human body. Mass loops are avoided, and any electric shocks or electrostatic discharges are prevented. In addition, for small amplitudes of the biological signal, electromagnetic interferences of below 100 Hz of the power supply network were eliminated by using an active fourth-order Bessel filtering module. The measurements performed for the low frequency of f = 100 Hz on the volunteers showed for the investigated organs that the bioelectrical resistivities vary from 90 Ωcm up to 450 Ωcm, and that these are in agreement with other published and disseminated results for each body zone.

## 1. Introduction

The measurement of bioimpedance is a technique for monitoring anatomical structure and physiological phenomena in the human body. In this regard, the measurement of bioimpedance can provide important information for applications such as: tissue hydration [1,2], body composition [1,3,4,5], vascular fluid flow [1,6], cancer detection [1,7,8], electrophysiology as electrocardiogram, electroencephalography [9,10,11,12,13,14], skin conductance, and hydration [14,15,16]. In addition, bioimpedance analysis has the advantages of being noninvasive, of low cost, good testing in the evaluation of health, and is safe and simple to perform [4]. Bioimpedance is a measure of the electrical properties of living tissues and the values of the electrical bioimpedance of tissues, as well as insight into intracellular and extracellular fluids, indicating the degree of their damage [8]. The basic principle of measuring the electrical bioimpedance consists of an injection of an alternating electric current of very low intensity into the biological human cells and tissues medium by means of two electrodes, and measuring the resulting electric potential difference by means of two other electrodes. The bioimpedance measurement is performed because the biologic medium behaves as a conductor, and at the same time as a dielectric or insulator of electrical current, which depends on the composition of the biological medium, thus resulting in a dependence of the bioimpedance values on the frequency. The insight of the electrical properties of the tissues into the intracellular and extracellular fluids can indicate the degree of tissue damage [14,15]. In biological media, three main dispersion regions can be associated with three ranges of values for frequency:The first range, α dispersion, for the frequency range of between 10 Hz to 10 kHz, is related to the phenomena of ionic diffusion of the cell membrane and the counterion effects;The second range, β dispersion, for the frequency range between 10 kHz and 100 MHz, is produced by the polarization phenomenon of cell membranes, the behavior of which is similar to that of capacitance. The polarization phenomena of proteins and other organic macromolecules contribute to the dispersion;The third range, γ dispersion, in the range of GHz, is produced by the polarization of water molecules [14,17,18,19,20].

A bioimpedance sensor must be capable of detecting impedance values for a certain tissue. In the case of a skin bioimpedance sensor, the sensor must be capable of measuring impedance values of between 200 Ω to 100 kΩ for a frequency range of 1 kHz to 100 kHz [15,16]. For an electrocardiogram, the frequency range is 1 Hz and 10 kHz [11,12]. In bioimpedance, measurements used several electrode configurations: with two, three, four, and five electrodes, [14,16,21]. For a two electrodes configuration, the voltage is measured at the same electrodes for the current injection. This configuration is mainly used for the monitoring of cell cultures via electrochemical impedance spectroscopy, where a particle disturbs the electric field lines, and therefore, the impedance, between two coplanar electrodes. Because the impedance of the electrode–medium interface is in series with the bioimpedance to be measured, errors may arise. To solve this problem, the four-electrode configuration (tetrapolar) is usually employed, where the current is injected via the external electrodes, and the resulting voltage is picked up by internal electrodes and applied to an amplifier [14]. For the bioimpedance measurements, the electrodes used are classified into polarizable and non-polarizable electrodes, wet or dry electrodes. For polarizable electrodes, no electrode reactions occur when an electrical potential is applied, and no charge flows across the electrode–electrolyte interface; the electrode behaves as a capacitor. For a non-polarizable electrode, the electrode will not change from its equilibrium potential in the presence of a density value of electric current that appears due to the injection of a constant current. For this type of electrode, the current flows freely across the electrode without producing any electric potential [21,22,23]. For the sensing potential, the polarizable electrodes are suitable, while for the current stimulation via current injection, the non-polarizable electrodes are suitable [21,24]. The dry polarizable electrodes are composed of metals as gold, stainless steel, carbon rubber smooth surface, carbon rubber textured surface, and conductive textile [25]. An alternative to these electrodes are the dry polarizable electrodes that are made of polymeric materials. Elastomers are the first class of polymeric materials that are used in bioimpedance electrodes for electrocardiogram monitoring; for example, PDMS (polydimethylsiloxane) filled with graphite or CNT (PDMS-black) [21,26,27,28]. Another class of polymeric materials from dry polarizable electrodes are represented by conductive polymers as such as polythiophene, polyaniline, and polypyrrole [21,29,30,31,32,33], and PEDOT/PSS and poly (3,4-ethylenedioxythiophene):(styrenesulfonic acid) [21,29,34]. An inexhaustive sample of applications under active investigation includes the identification of acute tissue damage [35,36,37] and skeletal muscle fatigue [38,39], the assessment of neuromuscular disorders [40], the monitoring of fluid movements during dialysis [41], the assessment of joint health [41,42,43], blood pressure monitoring [44], and respiratory monitoring [45].

The purpose of this study is to highlight a method of making equipment for the investigation of low frequency bioimpedance. A constant current with an average value of *I* = 100 µA is injected into the human body by means of the current injection electrodes E1 and E3, made from Ag (silver), and the biological signal charged with the biopotentials generated by the human body is taken from the electrodes E2 and E4, made from Au (gold). The mathematical model of the four-electrode system in contact with the skin and a target organ was simplified to a single equivalent impedance.

To make a quick comparison with the results obtained by other authors, the results of this study will be presented in terms of bioelectrical resistivity. The constant current injection used allows us this analogy, as shown in the paper. The measurements performed on seven volunteers show for the investigated organs that the bioelectrical resistivities vary from 90 Ωcm up to 450 Ωcm and are in agreement with other published and disseminated results for each body zone. The highest average resistivities, around 274 Ωcm for the breast/lung zone and 278 Ωcm for the liver zone, were obtained. The mean resistivity values were situated in the heart, kidney, and pancreas areas, these values were around 220 Ωcm. The lowest average bioelectrical resistivity resulting from the experiments was obtained for the right forearm (170 Ωcm), with 173 Ωcm for thyroid and 182 Ωcm for the left forearm zone. The liver zone measured with this equipment showed raw bioelectrical resistivity data lying between 200 and 450 Ωcm, these were the highest values and corresponded to the largest organ from human body.

## 2. Materials and Methods

### 2.1. Bioimpedance Measurement Sensors and Body Injection Current Electrodes

The bioimpedance measurement sensor is realized from four round disk shaped electrodes disposed at 6 cm one from another (Figure 1, Figure 2 and Figure 3). The sensing elements are made from two electrodes that are centrally positioned. The sensing disks are made from Au (gold). Each of the two electrodes that are used for current injection in the body are made from Ag (silver) and are positioned to the left and right, respectively, relative to the electrodes that make up the sensing elements. All of the electrodes have the same diameter and they are positioned at the same distance, 6 cm from one another. Electrical connections and contacts are made on the opposite side of the silver and gold disks, respectively.

All measurements have been carried out by using current injection electrodes made from Ag (silver), having impedances of Ze1 and Ze3, respectively, and the electric potentials of electrodes made from Au (gold), having impedances Ze2 and Ze4, as shown in Figure 1, Figure 2 and Figure 3.

The materials that are used to make the current injection electrodes are:-Ag (silver) foil, Code AG000470/37, having 99.99% purity, 0.5 mm thickness, purchased from ALDRICH;

The materials that are used to make the voltage sensing electrodes; named from now on, the electrodes of electric potential are:-Au (gold) foil, Code PREMION, having 99.99% purity, 0.127 mm thickness, purchased from ALFA AESAR.

The diameter (D) and thickness (t) of the electrodes that are made of Ag (silver) are D = 10 mm and t = 0.5 mm, and for the electrodes that are made of Au (gold), D = 10 mm and t = 0.127 mm. These geometrical dimensions were chosen in order to achieve a large skin contact surface and to surround with all electrodes the target organs surface. The current density corresponding to an injected current has a constant average value of *I*_1_ = 100 µA, which is identical in both cases, namely:(1)$$J_{Ag,Au}=\frac{I}{A}=\frac{100\cdot 10^{-6}\cdot 4}{\pi \cdot {\left(10\cdot 10^{-3}\right)}^{2}}=1.27\;A/m^{2}$$
where relation (1) determines the average current density.

### 2.2. General Cole Model for Skin, Organs, and Electrodes

The impedance of the skin and organs can be assimilated with a fractional capacitor arranged in parallel with a low frequency resistor *R*_0_ and a series contact resistance *R*_∞_ (gap) used for modeling high frequency circuits. The capacitor shapes the cell membrane, and in the same time, the corneous layer of the skin, which in the dry state can be considered as a dielectric, and in the wet state introduces a skin–electrode contact resistance in parallel. The parallel resistance can also indicate the amount of water or ionic electrolyte that is present in the targeted human organs. The cell membrane separates the extra and intracellular fluid, so that the bilayered fluid can be considered a dielectric. In addition to its passive role, the membrane contains ion channels and pumps that allow the ions to flow from one cell to another, forming gateways that are resistant to current carriers (Grimnes and Martinsen, 2008 [46]). Cell and organ membranes can be compared to multilayered wet electrolytic capacitors.

Thus, for the same reason, the cells and organ membranes are described here as two-plate capacitors *C_s_*, which are connected together with a parallel resistor *R*_0,*s*_–*R*_∞,*s*_, and a high frequency series resistor *R*_∞,*s*_; see Figure 4.

The model for the electrodes is structurally similar, containing a resistor (called a “contact gap resistor”) in series with a fractional impedance. In addition, there is a voltage Vs, which serves to simulate the conductive electrolyte layer that makes the ionic connection between the various layers of skin. The equivalent skin impedance (symbolized by s from “skin” in English) for the left circuit can be written:(2)$$Z_{s}\left(\omega \right)=R_{\infty ,s}+\frac{R_{0,s}-R_{\infty ,s}}{1+{\left(j\omega \right)}^{a_{s}}\cdot \left(R_{0,s}-R_{0,\infty }\right)C_{s}}$$
where *R*_∞,*s*_ is the contact (gap) resistor, so that *Z_s_*(∞) = *R*_∞,*s*_ for high frequency, *R*_0,*s*_ is the low frequency resistance, i.e., *Z_s_*(0) = *R*_0,*s*_, as is the fractional order and *C_s_* is the fractional electrical capacity of the skin or organ, [16,47,48].

At low frequencies, for f = 100 Hz, *R*_∞,*s*_ is at least two or three orders of magnitude smaller than *R*_0,*s*_, so that its contribution can be ignored for the parallel RC circuit, and Equation (2) becomes:(3)$$Z_{s}\left(\omega \right)=R_{\infty ,s}+\frac{R_{0,s}}{1+{\left(j\omega \right)}^{a_{s}}\cdot R_{0,s}C_{s}}$$

At 100 Hz or at a lower frequency, for example, *R*_0,*s*_ is around the order of MΩ value. In addition, *R*_∞,*s*_ is of the order of kΩ value, above and near a frequency of 40 kHz. Likewise, the equivalent impedance of an electrode (symbolized by e), for the right circuit can be written as:(4)$$Z_{e}\left(\omega \right)=R_{\infty ,e}+\frac{R_{0,e}-R_{\infty ,e}}{1+{\left(j\omega \right)}^{a_{e}}\cdot \left(R_{0,e}-R_{\infty ,e}\right)C_{e}}$$
where *R*_∞,*e*_ is the high frequency contact gap resistor, so we have *Z_el_*(∞) = *R*_∞,*e*_; *R*_0,*e*_ is the low frequency resistance, meaning that *Z_el_*(0) = *R*_0,*e*_, *a_e_* is the fractional order and Ce is the pseudo-capacitance (noted as CPE in other articles [47]). The mathematical model of an electrode is structurally identical to that of the Cole skin model [49], except for a c.c. potential of about 230–250 mV, added in series.

At low frequencies, for f = 100 Hz, the electrode contact resistance *R*_∞,*e*_ is at least two orders of magnitude smaller than *R*_0,*e*_, so that its contribution can be ignored for the parallel RC circuit and the equation becomes:(5)$$Z_{e}\left(\omega \right)=R_{\infty ,e}+\frac{R_{0,e}}{1+{\left(j\omega \right)}^{a_{e}}\cdot R_{0,e}C_{e}}$$

As we can see in the lower section, *R*_0,*e*_ is of the order of MΩ, and *R*_∞,*e*_ is under 10 kΩ. The Cole model was selected for this paper because it captures the behavior of arm muscles and organs with electrodes and sensors placed at short distances (usually 2–6 cm) and in a wide frequency range (from 1 Hz to 10 kHz or more), which covers some types of bio-signals.

As we can notice, the maximum distance of between 1 and 3 current injection electrodes is 18 cm, and the measuring distance between the sensing electrodes is 6 cm. That positioning distance was chosen to completely cover all organ areas. For medium sized organs such as the heart or kidney, the mean size is 12 cm, the brain is 18 cm, and the lungs and liver are the biggest organs, being up to 24 cm in size.

The stratum corneum skin layer has a predominant influence at lower than 1 kHz frequencies, when compared to the other skin layers, such as the epidermal, dermal and subcutaneous layers. The stratum corneum thickness varies between 10 and 20 μm where the bigger organs are positioned, and up to 0.1–2 mm on the palms, hands, and feet [50].

Bioimpedance at low frequencies is related to the resistive parts of the fluids, ions, and proteins outside the cell, but that does not mean that the capacitance of targeted organs can be neglected in this case. The skin should be regarded as one capacitance and one resistance mounted in parallel, and targeted organs should be regarded as another capacitance with parallel resistance. Practically, it is necessary that the two parallel R-C structures be considered as being mounted in series. The relative permittivity of the stratum corneum was between 1000 and 10,000 when the frequency was lower than 1 kHz; see Birgersson, Yamamoto et al. [50]. On the other hand, the measured permittivity of the liver was 82,300, plus or minus an error of 20,300 at 10 kHz [51]. If we consider the permittivity of the organs to be between 10,000 and 100,000, the skin influence on the measured capacitance should be very small, because the skin thickness (10–20 μm) is very small in comparison with an organ penetration depth of 30 cm at 100 Hz. In conclusion, at lower frequencies, between 100, …, 1000 Hz, we are able to distinguish between both the resistive and capacitive influence of the organs.

Alpha dispersion at a low frequency, from complex admittance Wessel plots, is usually very small, at between 0.0025 and 0.02 (10–100 Hz); see Sverre Grimnes et al. [52]. At lower than 50 Hz, frequencies admittance was often found to be equal to DC conductance. Like Sverre Grimnes et al. said, alpha dispersion has different values and should not be confused with the alpha exponent factor that appears in the Cole–Cole equations. In our paper, we have calculated only the alpha exponent factor by using a three data points method that is situated on a complex plane where the real and imaginary parts of the dielectric constant are represented. The dielectric constant model was represented by an arc of the circle where these data points are situated. From the loss tangent measurements that were situated between 0.5 and 0.7 and the capacitance measurements effectuated with the RLC bridge that were situated between 600 and 830 pF, we have managed to extract the real ε′ and imaginary ε″ parts of the permittivity dielectric constant. From each of the three data points measurements from a circle, where these real ε′ and ε″ imaginary parts of the permittivity dielectric constant must be positioned. Measurements with the RLC bridge were made in a 100–1000 Hz intermediate interval to obtain a reasonable accuracy.

After that, the dielectric constant, $\varepsilon_{0}$ at low frequency and $\varepsilon_{\infty }$ at high frequency, was calculated using formulas [53]:(6)$$\varepsilon_{0}=A+\sqrt{R^{2}-B^{2}}$$
(7)$$\varepsilon_{\infty }=A-\sqrt{R^{2}-B^{2}}$$

The A, B factors were depending on the three adjacent points situated on the dielectric constant circle of radius R; the data points were arranged in three groups of three, and the procedure was repeated until all the points were used. If we make the same measurements and calculations at low (100 Hz) and high (40,100 kHz) frequencies, we will observe that the model accuracy is decreasing. More about the A, B and R formulas and the calculation procedure are available in the article by Samiha T. Bishay [53].

The alpha exponent was estimated according to the equation:(8)$$\alpha =1-\frac{2}{\pi }\left({tan}^{-1}\left(\frac{\varepsilon^{\prime \prime }}{\varepsilon_{0}-\varepsilon^{\prime }}\right)+{tan}^{-1}\left(\frac{\varepsilon^{\prime \prime }}{\varepsilon^{\prime }-\varepsilon_{\infty }}\right)\right)$$
and has a value of *α* = 0.807. Because the capacitance was measured at 100 Hz, we have a higher relaxation time of *τ*_0_ = 0.0109 s.

The relaxation time was calculated with this formula:(9)$$\tau_{0}=\frac{1}{\omega }{\left[\frac{{\left(\varepsilon_{0}-\varepsilon^{\prime }\right)}^{2}+{\varepsilon^{\prime \prime }}^{2}}{{\left(\varepsilon^{\prime }-\varepsilon_{\infty }\right)}^{2}+{\varepsilon^{\prime \prime }}^{2}}\right]}^{1/\left(2\left(1-\alpha \right)\right)}$$

The relaxation time is usually situated between 10 to 100 microseconds for frequencies between 100 kHz and 1 kHz.

An useful feature of the Cole model is its time constant τ_s_, is defined as [47], and the relaxation time can also be defined as:(10)$$\tau_{s}=\sqrt[{a_{s}}]{\left(R_{0,s}-R_{\infty ,s}\right)C_{s}}$$

This relaxation time in Formula (10) has less accuracy when compared to the previous one.

The fractional order will be calculated by using a fitted data model, considering it compared to the measured impedances at low and high frequencies. The time constant τ_s_ could be described in another way by using the Debye permittivity model, and it is linked to the loss tangent tan (δ).

### 2.3. Modeling a System of Electrodes in Contact with the Skin and a Target Organ

By using the electrical circuits shown above for the skin and electrodes, a general wiring diagram for a four-electrode system in contact with the skin can be constructed. The current injection electrodes made from Ag (silver), having impedances of *Z_es_*_1_ and *Z_es_*_3_, respectively, (for each electrode, we are considering a corresponding impedance at the interface of the electrode with the skin) will inject a constantly in average value current of *I*_1_ = 100 μA into the human body (Figure 5). The electrodes of electric potential made from Au (gold), having impedances of *Z_es_*_2_ and *Z_es_*_4_, respectively, (for each electrode, a corresponding impedance is considered the interface of the electrode with the skin) will take over the electric potential from the body, usually between 1 and 20 mV (Figure 5).

The current injection electrodes hereinafter are referred to as 1 and 3, are made from Ag (silver), are identical in geometric structure, and are fabricated by using the same material. The electrodes of electric potential are hereinafter referred to as 2 and 4, are made from Au (gold), also have an identical geometric structure, and are fabricated by using the same material. Furthermore, the distances at which they are placed are equal to 6 cm, one from another. Therefore, we can consider that the total electrode–skin equivalent impedances of 1–2, 3–4, and 2–4 do not differ much from each other, which is observed practically from the measurements. Thus, it can be considered that *Z_es_*_1_ ≈ *Z_es_*_3_ and *Z_es_*_2_ ≈ *Z_es_*_4_. The capacitors *C*_23_ and *C_p_* from the differential amplifier entrances are in the order of 10 nF and 1 nF, respectively.

It can be written (Figure 6),
(11)$$I_{0}=\frac{U_{m}+2Z_{es2}I_{2}}{Z_{0}}=I_{1}-I_{2}$$

In the end, according to Thevenin’s theorem, the circuit can be simplified into a single *Z_esn_* equivalent impedance in parallel with the current source, and *I*_1_ can even be considered a short-circuit current. It is well known that the current source can also be regarded as a voltage source that will be shorted, and the same *Z_esn_* equivalent impedance will be mounted in series, as in Figure 6.
(12)$$Z_{esn}=2Z_{es2}+\frac{2Z_{es1}Z_{0}}{2Z_{es1}+Z_{0}}=2\left(Z_{es2}+\frac{Z_{es1}Z_{0}}{2Z_{es1}+Z_{0}}\right)$$

As mentioned, *Z_es_*_1_, *Z_es_*_2_, *Z_es_*_3_, and *Z_es_*_4_ are the bioimpedances corresponding to each electrode; more precisely, each is the corresponding impedance at the interface of the electrode with the skin. Finally,
(13)$$Z_{es1}=Z_{es3}=R_{es1}+\frac{R_{0,e1}}{1+{\left(j\omega \right)}^{a_{e1}}\cdot R_{0,e1}C_{e1}}+\frac{R_{0,s1}}{1+{\left(j\omega \right)}^{a_{s1}}\cdot R_{0,s1}C_{s1}}$$
(14)$$Z_{es2}=Z_{es4}=R_{es2}+\frac{R_{0,e2}}{1+{\left(j\omega \right)}^{a_{e2}}\cdot R_{0,e2}C_{e2}}+\frac{R_{0,s2}}{1+{\left(j\omega \right)}^{a_{s2}}\cdot R_{0,s2}C_{s2}}$$
(15)$$Z_{0}=R_{es24}+\frac{R_{0,s24}}{1+{\left(j\omega \right)}^{a_{s24}}\cdot R_{0,s24}C_{s24}}$$
where *Z*_0_ is the impedance of organ to be analyzed, *R_es_*_1_, *R_es_*_2_, *R_es_*_3_, and *R_es_*_4_ are the series resistances corresponding to each skin–electrode interface, and *R_es_*_24_ is the resistance corresponding to the organ at high frequency. *R*_0,*e*1_, *R*_0,*e*2_ *R*_0,*e*3_ R_0,*e*4_ and *R*_0,*s*1_, *R*_0,*s*2_ *R*_0,*s*3_
*R*_0,*s*4_, *R*_0,*s*24_ are the low frequency resistors for electrodes, skin, and organ (*R*_0,*s*24_). *R_es_*_1_, …, *R_es_*_4_ are skin–electrodes series resistances, and so both the skin resistance and the electrode resistance at higher frequencies are summed and represented by *R_es_*_1_, …, *R_es_*_4_. All other skin capacitive parts are also represented by *C_s_*_1_ and *C_s_*_2_. In addition, *C_e_*_1_, *C_e_*_2_, *C_e_*_3_, *C_e_*_4_ and *C_s_*_1_, *C_s_*_2_, *C_s_*_3_, *C_s_*_4_, *C_s_*_24_ are the electrode, skin, and organ (*C_s_*_24_) capacitances, which can be regarded as a group of three series mounted capacitances with different dielectrics. The current will flow only between two electrodes; thus, there are two branches in parallel with the target organ, one for the body current injection and one for voltage sensing. The fractional orders *a_e_*_1_, *a_e_*_2_, *a_s_*_1_, *a_s_*_2_, and *a_s_*_24_ are usually determined from a fitted data model, when compared to the real obtained impedances at low and high frequencies. Usually, a fractional order value is in the range of [0.6, 0.97] [49]. From a theoretical point of view, *C_e_*_1_ ≈ *C_e_*_3_, *C_e_*_2_ ≈ *C_e_*_4_, *C_s_*_1_ ≈ *C_s_*_3_, *C_s_*_2_ ≈ *C_s_*_4_, *R_es_*_1_ ≈ *R_es_*_3_, *R_es_*_2_ ≈ *R_es_*_4_ and *R*_0,*e*1_ ≈ *R*_0,*e*3_, *R*_0,*e*2_ ≈ *R*_0,*e*4_, if the skin and organ parts are similar. In practice, the skin and organ parts are a little bit different and the applied pressure on each electrode is another issue. Because the contact surface for each electrode is not identical and the body surface is not the same in all directions, we obtain slightly different values for capacitors and resistances. Overall, the total equivalent impedances are maintained in the same measuring range, and the simplified model should be valid. Several measurements were made at a frequency of 100 Hz to determine the capacity and series resistance (*C_s_*, *R_s_*) depending on the organs analyzed. The same thing was performed to determine the capacity and resistance in parallel (*C_p_*, *R*_0*p*_).

Three separate cases were analyzed, depending on the placing of the electrodes. In the first case, electrodes 1 (Ag) and 2 (Au) have been placed in the target areas such as the forearm, intestine, and liver (Table 1). In the second case, electrodes 3 (Ag) and 4 (Au) have been placed in the same target areas such as the forearm, intestine, and liver (Table 2). The third case was obtained by placing electrodes 2 (Au) and 4 (Au) in the same target areas such as the forearm, intestine, and liver, Table 3.

The equivalent bioimpedance, if the parallel *R*_0*p*_-*C*_*p*_ scheme is considered, can be calculated as:(16)$$Z_{peq}=\frac{1}{\sqrt{{\left(\frac{1}{R_{0p}}\right)}^{2}+{\left(\omega C_{p}\right)}^{2}}}$$

The four electrodes positioning system from Figure 7 consists of a mechanical device for constant pressure that is simply a hard rubber support and four small soft rubbers that act like springs when attached to the back side of each electrode. The measurements with two electrodes for which the results in Table 1, Table 2 and Table 3 are presented were carried out to verify if *C_e_*_1_ ≈ *C_e_*_3_, *C_e_*_2_ ≈ *C_e_*_4_, *C_s_*_1_ ≈ *C_s_*_3_, *C_s_*_2_ ≈ *C_s_*_4_, *R_es_*_1_ ≈ *R_es_*_3_, *R_es_*_2_ ≈ *R_es_*_4,_ *R*_0,*e*1_ ≈ *R*_0,*e*3_, *R*_0,*e*2_ ≈ *R*_0,*e*4_ and *Z_es_*_1_ ≈ *Z_es_*_3_, *Z_es_*_2_ ≈ *Z_es_*_4_, respectively, such as we have mentioned in the previous modeling chapter.

The polypyrrole, composite 10% Ag/PPY as a sensitive material, Telipan, G. et al. [21] had a total measured impedance of between 350 and 200 Ω, and the parallel resistance is in the same range; thus, *R*_∞_ is in the range of 200–400 Ω, because the applied frequency was between 10 and 300 kHz. The measurements with polymer electrodes were made directly with the RLC bridge specified in the paper; see also [21].

The carbon rubber electrodes are used in ECG, EMG or muscle stimulation measurements to a frequency of below 1 kHz. The carbon rubber electrodes can be applied directly to the skin, and they can adapt different shapes because of their elasticity [25,54].

The metal electrodes such as gold (Au) are the best choice for their use in low frequency bioimpedance instrumentation [25]. In addition, AgNWs (Ag nanowires) are metallic dry electrodes used in bioimpedance with good performance in electrophysiological sensing as ECG and EMG measurements. For the subjects, ECG testing the signals measured with the AgNW electrodes are comparable with those obtained with wet Ag/AgCl electrodes. In addition, the AgNW presents a series of advantages against Ag/AgCl electrodes; namely, no skin irritation, as well as signal degradation for long-term wear, and less motion artifacts. Because of the good performance in electrophysiological sensing, the AgNW electrodes can be a good alternative of conventional wet Ag/AgCl electrodes. However, for long term use, the AgNW electrodes may oxidize, and for the elimination of this inconvenience, the AgNW electrodes can be embedded in an elastomer layer [55].

The electrodes of electric potential as sensing elements 2 and 4 made from Au (gold) were measured at high frequencies, and we have seen a variation from 2.6 kΩ down to 426 Ω for the series resistances *R*_∞_; the corresponding frequency was set up from 40 kHz up to 300 kHz. We can conclude from those measurements that the high frequency series resistances *R*_∞,*e*_ or *R*_∞,*s*_, for these electrodes and for skin, does not exceed the 1 kΩ limit for frequencies higher than 100 kHz (Table 4).

As we have determined at 100 Hz, the sensor-body series resistances *R*_*es*24_ at high frequencies are not equal for the forearm, large intestine, and liver zones.

To be noted again that the resistance and capacitance in parallel changes depending on the pressure exerted to achieve the best possible mechanical contact with the skin, the maximum capacity in parallel will not exceed 5 nF at 100 Hz and the minimum resistance will not fall below 400 kΩ. For the performed measurements, a relatively constant average pressure was considered. Even if the parallel capacity variation can be between 600 and 900 pF, the measured parallel resistance tends to compensate, increase, or decrease slightly, in order to finally obtain about the same equivalent bioimpedance, corresponding to the analyzed body area.

With the RLC bridge, we have analyzed three areas of the body at a frequency of 100 Hz, the area of the forearm, the area of the large intestine, and the part where the liver is positioned. The tables clearly show a variation of the equivalent bioimpedance, depending on the analyzed body area. The total series impedance between electrodes 1 (Ag) and 2 (Au) mounted on the skin is around 1.33 MΩ in the forearm area; this decreases slightly to 1 MΩ in the large intestine area and decreases even more, to 787 kΩ, in the liver area. The total parallel impedance for electrodes 1 (Ag) and 2 (Au) mounted on the skin behaves exactly the opposite, increasing from 1.377 MΩ in the forearm area, to 1.4 MΩ in the large intestine area, and to 1.9 MΩ in the liver area.

Thus, it can be said that there is a clear correlation between the value given by the device, which displays a higher average value of resistivity in the liver area and an average value about twice lower in the forearm area. The area of the intestine has an average value of resistivity just below that in the zone of the liver. From the RLC bridge measurements, it seems that at low frequencies, we cannot distinguish between the skin and organ impedances; overall, we obtain the same *Z*_0_ ≈ *Z_es_*_3_ ≈ *Z_es_*_2_ impedance values. At a low frequency, 100 Hz, we cannot precisely visualize inside the layers of the skin and inside the organ membranes; we obtain only a rough, general view of the entire targeted area.

The laboratory equipment used for all the experiments are the following:

Agilent E4980 A LCR Meter;

Keysight 53,210 A RF Counter;

FLUKE 281 40 Ms/s Arbitrary Waveform Generator;

Agilent 34,461 A 6 ½ Digit Multimeter;

Tektronix TDS 2014 B Digital Storage Oscilloscope;

NI LabVIEW Signal Express Tektronix Edition.

Measurements were made at room temperature (22 ± 3 °C) and with humidity 48.2 RH.

## 3. Design of Equipment for the Investigation of Low Frequency Bioimpedance

### 3.1. Electronic Conditioning System Used in Biological Signals Acquisition

The electronic conditioning system used in biological signals acquisition and monitoring is composed according to Figure 7, from:-Module 1—electronic module for acquiring biological signals;-Module 2—filtering electronic module;-Module 3—common mode amplifier module for electronic signal output.

Constructively, the electronic conditioning system is made by cascading the electronic modules Module 1, Module 2, and Module 3. Functionally, the biological signals received from the electrodes E2 and E4 charged with biopotentials from the human body, are extracted using an instrumentation amplifier IC 1, type INA 111, fabricated by Burr-Brown, from Texas Instruments, and are used in differential input connections; see Figure 7 and Figure 8. A precision amplifier with a differential capacitive isolated barrier IC 2 type ISO 124, manufactured by Burr-Brown, Texas Instruments, picks up the signal resulting from the IC 1 output (Figure 7 and Figure 8). The signal resulting from the output of the capacitive isolated precision amplifier IC 2 is then applied to an active Bessel band-pass filter of order four (Figure 9 and Figure 10), made with the operational amplifiers IC3 and IC 4, OPA 134 type, fabricated by Burr-Brown, Texas Instruments. The filter parameters are: a center frequency f = 100 Hz, cutoff frequency attenuation −3 dB, stopband attenuation of −45 dB, and a 1 Hz band-pass frequency range. Figure 11 shows the characteristics of the active Bessel band-pass filter of order four, drawn based on laboratory measurements data.

The signal resulting from the output of the active Bessel band-pass filter of order four is then amplified by the electronic amplifier module of common output mode, made with operational amplifiers IC5, IC6, and IC7, LF 356 type, manufactured by Texas Instruments (Figure 12).

#### 3.1.1. Electronic Module for Acquiring Biological Signals

The electronic module for receiving biological signals taken from the electrodes charged with biopotentials generated by the human body (Figure 8) is made using an instrumentation amplifier, IC 1, with a very high common mode rejection ratio CMRR; at least 106 dB. In addition, bias currents of up to 20 pA in input circuits are realized with FET transistors, which are used in the “differential input mode” connection to be able to accurately pick up biological signals, and then using a precision amplifier with IC 2 galvanic isolation to further remove unwanted interferences.

Setting the value of the amplification factor for the instrumentation amplifier IC 1 to A = 100 is performed by adjusting the external resistor R_G_ (Figure 8), to the value R_G_ = 505.1 Ω.

A low pass filter mounted at the IC 1 input is made with capacitors C_in1_, C_in2_, and C_in3_. The values of these capacitors are found in the relation C_in1_ = C_in2_ and C_in3_ = 10 C_in1_, so as to improve the CMRR. The input impedance of the IC 1 amplifier is extremely high, approximately 1012 TΩ, with the purpose of minimizing the distortion of the detected biological signal. By cascading the precision amplifier IC 1, with a galvanic capacitive isolated precision amplifier IC 2, having a nominal breakdown voltage rating of 1500 V rms value, avoids mass loops, and independent processing of the signal occurs, resulting the isolating of output of the amplifier. At the same time, the prevention of electrical shocks to the investigated human subjects is realized.

The IC 2 integrated circuit is a precision isolated amplifier that incorporates a modulation-demodulation technique, and the signal is transmitted digitally through a two pF differential capacitive barrier. With digital modulation, the characteristics of the barrier do not affect the integrity of the signal. The key specifications are a maximum nonlinearity of 0.010%, the signal bandwidth is 50 kHz, and the V_OS_ drift = 200 μV/°C.

The power supply of the electronic module for acquiring the biological signals (Figure 8 and Figure 9), is made with a protected double differential isolated voltage source. The first differential source supplies the IC 1 instrumentation amplifier and the IC 2 input section, and the second differential source supplies the IC 2 output section, which is galvanically isolated from the input section. Each of these differential power supplies are separated one from another, and each of these masses are separated and are in relation to which the reference potentials are considered, with the help of the capacitors C1, C3, and C11; and C2, C4, C12, C5, and C6, respectively.

The PCB design adopted the DIP package version of all chips to quickly possess functional prototypes for each desired fixed frequency of 100 Hz, 40 kHz, and 1 kHz. Please keep in mind that we have designed and implemented band-pass filters with very narrow bandwidth ranges (1 Hz for 100 Hz center frequency) for each fixed frequency in order to reject all other unwanted interfering signals and to keep only the useful signal.

A smart multi-frequency bioelectrical impedance spectrometer that uses the AD5933, AFE4300, AD5940, or MAX30003 chip will have some difficulties in using very narrow band-pass filters; separate low pass and high pass filters will be a more suitable choice in this case.

Although the INA111 instrumentation amplifier is an old chip that has been produced since 1998, with the MAX30003 and AD5940 microchips having been newer options since 2019, it provides a 100 times amplification of the bioelectrical signals, from 1–20 mV up to 0.1–2 V, ensuring a higher accuracy and sensitivity, and together with the isolation amplifier ISO124, a higher immunity to electromagnetic noise and accidental voltage spikes (up to 1500 V).

This paper [36] observes that the internal high pass filters of MAX3000x have a negative impact on measurements, and recommends that the analog high-pass filter should remain disabled for frequencies < 80 kHz. The error decreased from 20% down to 4% when the programmable gain was maximized for MAX3000x at higher frequencies. For this chip, a better accuracy was clearly obtained when the gain was set to minimum (10) and the frequency was below 40 kHz, and when the gain was set to maximum (80), the best accuracy was obtained for higher than >40 kHz frequencies.

One drawback of a fixed frequency bioimpedance device is that it cannot establish in real time any fluid or fat changes. The alpha dispersion at low frequency was very small, at between 0.0025 and 0.02 (10–100 Hz), and the Wessel plots [52] of complex admittance showed little difference; at lower frequencies, and the admittance was often found to be equal to the DC conductance.

#### 3.1.2. Electronic Filtering Module

The electronic filter module (Figure 10), is an active fourth-order Bessel band-pass filter, and it is made with the use of the operational amplifiers IC3 and IC4, with a very low distortion of 0.00008% and a low noise of 8 nV/√Hz. The cascaded input circuits of the operational amplifiers IC3 and IC4 ensure an excellent rejection ratio of the common mode and keep the low bias current at the input of I_B_ = 5 pA over a wide range of input voltages, minimizing distortions. The offset voltage can be adjusted by connecting a semi-adjustable potentiometer between pins 1 and 8, and the cursor of this potentiometer is connected to the positive potential + V_S2_, as shown in Figure 10.

Filter Pro software from Texas Instruments was used to generate the topic of the electronic filter module, as well as the values of all components.

The main features of the fourth-order Bessel band-pass filter are: center frequency f = 100 Hz, cutoff frequency attenuation of −3 dB, stopband attenuation of −45 dB and 1 Hz bandwidth range.

Figure 11 shows the graphical representation of the main characteristic of the fourth-order active Bessel band-pass filter, realized based on laboratory measurements. Due to the fact that the real operational amplifiers have different characteristics from the ideal operational amplifiers with which the Filter Pro software program operates, the frequency of the real fourth-order active Bessel band-pass filter is centered on the value of f = 102 Hz (Figure 11).

#### 3.1.3. Electronic Common Mode Output Amplifier Module

The electronic common mode output amplifier module (Figure 12), is made with three operational amplifiers, IC5, IC6, and IC7, with very good dynamic performance, i.e., fast Slew Rate = 12 V/µs. These operational amplifiers are used in the inverting configuration, cascaded series connections, and with the signal pickup mode in common-mode connection, to take the useful signal from the electronic filtering module and to amplify it to the desired total amplification factor. The useful signal refers each time to the signal on the signal path of the electronic conditioning blocks.

The individual amplification factor of the operational amplifiers IC5, IC6, and IC7 is A = 4.44. The global amplification factor is the product of the amplification factors, i.e., A_T_ ~ 87.5.

The offset voltage can be compensated using the semi-adjustable potentiometer R_OFS_ = 25 kΩ (Figure 12) through an iterative process. To this effect, each inverter input of the operational amplifiers IC5, IC6, and IC7 is connected consecutively to the reference potential, and the semi-adjustable potentiometer R_OFS_ is adjusted until zero volts are obtained at the output of each operational amplifier.

Figure 13 shows a capture of the waveform from the output of the electronic conditioning system for use in acquiring biological signals, in case of application at the input of a sinusoidal signal with the characteristics: U_peak-to-peak_ = 10 mV and f = 100 Hz. Figure 14 shows a capture of the waveform from the output of the electronic conditioning system for use in receiving biological signals, in the case of an application at the input of a sinusoidal signal with the characteristics: U_peak-to-peak_ = 12.5 mV and f = 100 Hz. It can be seen that the resulted signal remains clear and undistorted; having a final peak-to-peak voltage of 19.8 V from a 10 mV peak-to-peak voltage input signal, and respectively, a peak-to-peak voltage of 25.8 V from a 12.5 mV peak-to-peak voltage input signal.

Figure 15 shows the electronic conditioning system used for acquiring and monitoring the biological signals, practical realization.

### 3.2. Electronic Module of the Constant Current Generator, I1 = 100 μA Average Value

The constant current with an average value of *I* = 100 µA is injected into the human body by means of the current injection electrodes E1 and E3 (Figure 8). The biological signals are acquired with the help of electrodes E2 and E4 charged with biopotentials from the human body, and the resulting voltage, Δ*U*, which represents the useful signal, is processed by the electronic conditioning system. The value of this voltage is in the range Δ*U* = 5–50 mV.

The electronic module of the constant current generator, of an average value of *I* = 100 µA, is made using the following electronic blocks:Electronic sinusoidal oscillator block;Buffer electronic block;Electronic block of voltage/constant current converter, average values of 100 µA (Figure 16).

The sinusoidal electronic oscillator block with parameters f = 100 Hz, A_vv_ = 3.2 V is made with the electronic function generator chip 8038, and an additional printed circuit board (Figure 17a,b). A function generator, sometimes also called a waveform generator, is a device or an electronic circuit that produces a variety of different waveforms at a desired frequency. The function generator can generate sine waves, square waves, or triangular waveforms. One such device is the 8038 circuit, an electronic precision waveform generator that is capable of producing sinusoidal, rectangular, and triangular waveforms by using a minimum number of external components or settings.

Its operating frequency range can be selected over eight decades of frequency, from 0.001 Hz to 300 kHz, by choosing the correct external R–C components. These components are chosen so that the frequency of the sine waves can be adjusted to f = 100 Hz.

An electronic buffer module consisting of a pair of complementary bipolar transistors, T1 and T2, is inserted between the electronic block of the sinusoidal oscillator, and the electronic block of the voltage/constant current converter of 100 µA in average value (Figure 16). This buffer circuit acts like a low impedance voltage source.

The electronic voltage/constant current converter block, with 100 µA in average value, is an electronic block for which the load current does not depend on the load value. This converter was made using the LF 356 operational amplifier, manufactured by Texas Instruments, in the configuration presented in Figure 16 and Figure 17. Considering the LF 356 operational amplifier an ideal operational amplifier, see Figure 17, it can be written:*V*_−_ = *V*_+_(17)

However, *V*_+_ = *V_IN_* (*t*), that is, a fraction of the voltage collected by means of the potentiometer P1 with a value of 1.5 kΩ, from the output of the sinusoidal oscillator. Using Kirchhoff’s laws for the branch through which the load current flows, we obtain:*V_DD_* − *R_SENSE_ I_OUT_* = *V*_−_(18)
as *V*_−_ = *V*_+_ = *V_IN_* (*t*), the equation above can be written as:*V_DD_* − *R_SENSE_ I_OUT_* = *V_IN_* (*t*)(19)
from where it is obtained:(20)$$I_{OUT}=\frac{V_{DD}-V_{IN}\left(t\right)}{R_{SENSE}}$$

U_GS_ voltage of the MOS P CHANNEL IRF 9540 transistor (Figure 17) is stabilized by the resistance R1 of 35 kΩ. Since it was necessary to achieve a constant current of 100 µA in average value, we can write:(21)$$I_{OUT}\sqrt{2}=0.0001\sqrt{2}=\frac{15-V_{IN}\left(t\right)}{R_{SENSE}}$$

Considering the input voltage at the value of *V_IN_* = U_PP_ = 3.12 V, it is obtained: *R_SENSE_* = 83.438 kΩ. The constant current, having *I* = 100 µA, it is considered as an average value (RMS). A constant current source was realized with a P-channel MOSFET and an LF356 operational amplifier to achieve a higher precision and better stabilized current (100 µA) by adjusting the feedback resistor P2 = *R_SENSE_* to 83 kΩ, and by considering the input reference voltage to be 3.12 V. This error amplifier will subtract this voltage fraction (3.12 V) collected by the 1.5 kOhm potentiometer P1 (Figure 16) from the +15 V supply voltage. If we divide the resulting voltage to 83 kΩ and multiply it by a square root of two, we obtain an average constant output current of about 100 µA that passes between the attached skin electrodes (E1 and E3). Since this 3.12 V voltage supply fraction is much lower than +15 V, the P-MOSFET transistor will always be in the saturation region where the output current does not vary with changes in the output voltage.

Figure 16 shows the electronic diagram of the constant current generator electronic module, having an average value of *I* = 100 µA.

In Figure 18, the characteristic of the constant current generator is shown with an average value of *I*_1_ = 100 μA, when the current is depending on the load. It is observed that for a load up to *Z* = 17.5 kΩ, the current does not fall below the value of *I*_1_ = 100 μA.

Seeing that:(22)$$\rho_{peq}=k\frac{\Delta U}{I}$$
and
(23)$$Z_{peq}=\frac{\Delta U}{I}$$
were

-*k* is a constant;-∆*U* is the biological signal charged with the biopotentials generated by the human body that is taken from the electrodes E2 and E4 (Figure 8);-*I* is a constant current in the average value, with a peak value of *I* = 100 µA that is injected into the human body by means of the current injection electrodes E1 and E3 (Figure 8). If we consider the four-electrode configuration (tetrapolar), where the external electrodes inject a constant current, Gómez-Cortés, J.C. et al. as well as Yu, Y. et al. [56,57] and the resulting voltage is picked up by internal electrodes and applied to an amplifier, it can be considered that the two physical quantities, bioimpedance and bioelectrical resistivity, differ only by a constant.

## 4. Experimental Data Obtained with Equipment for the Investigation of Low Frequency Bioimpedance and Evaluation of the Accuracy of the Measurements

Experiments were performed by using seven human volunteers selected from a research institute. For only two of them, the device showed bioimpedance values as being higher than normal, in the targeted zones. We presumed that this was because their body proportions were different from the average persons, but more studies need to be performed in that direction.

The mean values of body bioelectrical resistivity obtained for different organs (Figure 19) from seven volunteers, were compared to the study by T.J.C. Faes, H.A. van der Meij and others [48]. The highest average resistivity calculated values from this earlier paper [48] were 339 Ωcm for the breast zone and 342 Ωcm for the liver. These values were also the highest values that were measured with the equipment for the investigation of low frequency bioimpedance, i.e., around 274 Ωcm for the breast/lung zone and 278 Ωcm for the liver zone (Figure 19). The obtained mean resistivity values were situated for the heart, kidney, and pancreas; these values were around 220 Ωcm. When the obtained data are compared to the data of T.J.C. Faes, [48], i.e., 175 Ωcm for heart and 211 Ωcm for the kidney, we can observe that these values are pretty close. The lowest average bioelectrical resistivity (Figure 19) resulting from these experiments was obtained for the right forearm, with 170 Ωcm and 173 Ωcm for thyroid and 182 Ωcm for the left forearm zone, which was also comparable with the results obtained from [49]. When we take a look over T.J.C. Faes, [48], and Freeborn, T.J., [49], we see that a lower resistivity of 171 Ωcm was calculated for the muscles, 183 Ωcm for the thyroid, and a lower impedance of between 100 and 200 Ω for both the arms and the legs. The device amplification voltage was adjusted to display the body resistivity value in Ωcm. In this way, we can neglect the electrode geometry and positioning distance, and the device is easier to calibrate.

From the experimentally obtained data, we can see in Figure 20 that both the right and left arms have a measured bioelectrical resistivity of between 80 and 400 Ωcm.

It is obvious that we will have a different bioelectrical resistivity, as the arms have different proportions because of the muscle strength, deposited fat, or different bone densities. Bones and fat are considered to be non-conductive, and they will have a much higher impedance when compared to the cell and organ membranes that are more responsible for conductance or low electrical resistances. Some study participants, although they are skinny, could have bones with a higher mineral density, as we see in Figure 20, subject number 5. Some will have more deposited fat; see subject number 7 in Figure 20, Figure 21 and Figure 22. All of the measurements are higher than the average in this case.

Measurements performed in the liver area (Figure 22) showed raw bioelectrical resistivity data lying between 200 and 400 Ωcm. It is interesting to see that the minimum obtained value is 200 Ωcm, which somewhat proves that this is one of the largest organs in our body.

Non-alcoholic fatty liver disease can be discovered due to a higher-than-average liver bioimpedance. In conclusion, low frequency measurements of between (100 Hz and 10 kHz) are proving to be useful for the health monitoring of various organs, local body water absorption, body fat deposits monitoring, and bone density quantitative estimations.

We have simulated and measured bioelectrical resistivity for different organs at frequency of 100 Hz (Table 5), and we have concluded that a resistance of between 6.172 and 6.179 kΩ can represent the resistive parts of the skin fluids and ions outside the cells. Skin capacitance was considered to be inside a 1 and 5 nF interval; here, this was chosen to be around 1 nF for all calibration circuits. The simulated capacitance *C*_0_ of the targeted organs was selected; after multiple trials, experiments, and comparisons, it was between 220 and 222 nF. This value is much higher if it is reporting the skin capacitance. The difference between the series capacitances of the skin and organs seems obvious because of the relative permittivity of the stratum corneum that can be even 100 times lower than the measured permittivity of the liver, e.g., 80,000, …, 100,000, at 10 kHz [51]. In addition, the penetration depth for organs is approximate 30 cm, and the skin thickness around the organs is 10–20 microns, so the equivalent permittivity of these two mixed dielectrics, skin and organ, remains to be approximately equal to those of the targeted organ.

By using the values results and summarizing in Table 5, we have managed to simulate four targeted body areas, namely, the forearm, kidney, large intestine, and liver zones, with only RC parallel circuits arranged as in Figure 6 or Figure 23, respectively. The impedances *Z*_*es*1_ and *Z*_*es*3_ are almost equal. Thus, electrode–skin impedances are represented by the passive components *R*_1_ and *C*_1_ mounted in parallel (Figure 23). The impedances *Z*_*es*2_ and *Z*_*es*4_ sensing electrode–skin measure almost equal impedances. These impedances are made of *R*_2_ and *C*_2_ passive components mounted in parallel (Figure 23). The selected organ impedance *Z*_0_ is constructed from the resistance *R*_0_ and the capacitance *C*_0_ mounted in parallel (Figure 23).

The bioelectrical impedance analyzer (Figure 24) was calibrated by comparing the testing circuit attached to the 1, 3, and 2, 4 terminals (Figure 23), made of parallel RC passive components with the mean resistivity values of the forearm, large intestine, and liver obtained directly from the measurements performed on human subjects and then by comparing with the T.J.C. Faes paper [48]. We may note that the values that presented in the T.J.C. Faes paper [48] have a 95% confidence interval.

The values of the *Z*_*es*1_ and *Z*_*es*3_ impedances are both 6.172 kΩ, and the values of the *Z*_*es*2_ and *Z*_*es*4_ impedances are both 6.179 kΩ. Using equation (12), the equivalent measured and also calculated *Z_esn_* impedance of the entire circuit was varied between 16,450 and 16,470 Ω, depending on the targeted organ capacitances (Table 5).

## 5. Discussion

This equipment was designed for the frequency of f = 100 Hz and was manufactured for the bioimpedance monitoring of all organs. The bioimpedance measurement sensors are realized from four round disk shaped electrodes, placed 6 cm one from another, to achieve a large skin contact surface and to surround with all electrodes the target organ surfaces. The sensing elements are made from two electrodes that are the disk shapes, and they are made from Au (gold) and are centrally positioned to pick up the electric potential from the selected organ. The silver electrodes are used for a 100 µA constant current injection. Each of the two electrodes that are used for the current injection in the body are made from Ag (silver) and are positioned to the left and right, respectively, relative to the electrodes that make up the sensing elements. Electrical connections are made on the opposite sides of the silver and gold disks, respectively. The mechanical device for supporting the electrodes with constant pressure was simply a hard rubber support and four small soft rubbers that act like springs when attached to the back side.

The mathematical modeling of a system of four electrodes in contact with the skin, placed at equal distances, and with identical shapes two of by two, was simplified by considering that for similar electrodes and skin zones, we have equal impedances, and the electric circuit was reduced to the single equivalent impedance.

For small signal amplitudes of 5–10 mV, the electromagnetic noise can distort the useful signal until the resulted signal from the output becomes unrecognizable, and thus, incorrect results can be obtained. However, the way in which this equipment was designed and realized for the investigation of low frequency bioimpedance removes this deficiency.

The supply network is responsible for 50 Hz electromagnetic interference, but other electromagnetic signals situated under 100 Hz could appear. The human body could be locally charged and be able to produce small electric discharges when the sensors are attached. These events are eliminated by using a low pass capacitive filter at the input, together with a capacitive isolated barrier integrated in the secondary precision amplifier and a fourth-order Bessel band-pass filter type. This active filtering module is mounted between a precision isolated amplifier and a common mode output amplifier module. The band-pass Bessel filter was designed to eliminate all frequencies at around 99 and 103 Hz, keeping only the useful 100–102 Hz signals used for current injection.

The biological signals received from electrodes E2 and E4, charged with biopotentials from the human body, are extracted by an instrumentation amplifier that is used in differential input connection. The mass from this first instrumentation amplifier is completely separated from the secondary precision amplifier mass that has a capacitive isolated barrier; in this way, ground loops are avoided and the measuring signal is independently processed, remaining clear and undistorted.

The third common mode output amplifier module is constructed from three cascaded operational amplifiers and has a high global amplification factor of 87.5. These amplifiers have a very good dynamic performance and a Fast Slew Rate = 12 V/µs, and are used in inverting the configuration to take the useful signal from the electronic filtering module and to amplify it to the desired amplitude. Thus, a final peak-to-peak voltage of 19.8 V from a 10 mV peak-to-peak voltage input signal was achieved, and respectively, a peak-to-peak voltage of 25.8 V from a 12.5 mV peak-to-peak voltage input signal.

The measurements performed on seven volunteers showed for the investigated organs that the bioelectrical resistivities vary from 90 Ωcm up to 450 Ωcm and are in agreement with other published and disseminated results for each body zone. Bioimpedance measurements were made for different body shapes, skinny, normal, and fatty, and different body compositions were analyzed that had fluid, mineral, and fat changes.

The highest average resistivities of around 274 Ωcm for the breast/lung zone and 278 Ωcm for the liver zone were obtained. The mean resistivity values were situated in the heart, kidney, and pancreas areas; these values were around 220 Ωcm. The lowest average bioelectrical resistivity resulting from the experiments, 170 Ωcm, was obtained for the right forearm, and 173 Ωcm for thyroid and 182 Ωcm for the left forearm zone.

The liver zone measured with this equipment showed raw bioelectrical resistivity data lying between 200 and 450 Ωcm; these are the highest values that correspond to the largest organ from the human body. For best results, this device needs to be calibrated further, and the experiments must continue on a much larger number of volunteers to determine a clear difference between the healthy organs and some diseased fatty organs. The low frequency, namely f = 100 Hz, was used so that the penetration depth of the constant current at an average value was approximately 30 cm, and thus, information about the internal organs could be received. The influence of the skin for low frequencies, as below 100 Hz, can be neglected. The skin impedance can be highlighted only for a working frequency of approximately 40,000 Hz, the frequency for which the penetration depth is very small, and the current is distributed only on the surface of the skin [21], so that at this time the influence of the internal organs is insignificant. If we increase the frequency to above 10 kHz, the tissue response will become dominant and we will analyze only a small portion of the targeted organ. Thus, we will perform further experiments and choose the right frequency situated between 100 Hz and 10 kHz until we obtain the best results for all targeted body zones.

The presented equipment proposed the investigation of bioimpedance at low frequency, namely for f = 100 Hz. The mathematical model of the system with four electrodes in contact with the skin and considering a target organ, represented a starting point in the development of the equipment for the investigation of bioimpedance at low frequency. As has been presented, the proposed equipment includes a current generator with a frequency of f = 100 Hz, but with a constant average value of 100 µA. This current is injected into the human body by means of current injection electrodes, and is placed in the area of a target organ. The resulting biological signal is picked up by the electrical potential electrodes, which will thus be charged with the biopotentials generated by the human body corresponding to the target organ. The biological signal corresponding to the target organ, i.e., the resulting voltage, is applied to the electronic conditioning system. Many impedance measurement circuits proposed in the literature cover a wide range of frequencies [58]. However, investigating the impedance for a single frequency, in our case, f = 100 Hz, has multiple advantages, such as:(a)The possibility of using a fourth-order band-pass filter with a very narrow bandwidth, even of 1 Hz (Figure 10 and Figure 11);(b)The possibility of picking up some biological signals from the target organs having extremely small amplitudes, up to 5 mV, precisely as a consequence of using the very narrow bandwidth filter;(c)Increasing the measurement precision by placing an additional filter in the cascade, a Bessel fourth-order band-pass filter, also having a very narrow bandwidth of 1 Hz.

More than that, the electronic conditioning system used in order to take over the biological signals from the target organs by means of the electric potential electrodes has an important particularity, namely: a precision amplifier with isolation is used, powered by stabilized differential sources with isolation (Figure 8). This approach is very useful in medical applications, as it can prevent electric shocks that human subjects could suffer during investigations. In order to achieve additional protection, the authors designed a protection system, Pîslaru-Dănescu, L. et al. [59], able to remove the bioimpedance analyzer from under the supply voltage. This protection system is applicable to any device in the medical field.

## Figures and Tables

**Figure 1 micromachines-13-01858-f001:**
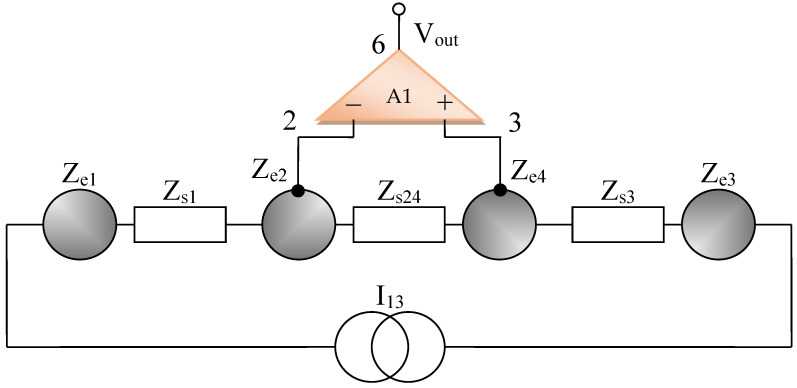
Simplified wiring diagram for the bioimpedance sensor, electrodes layout.

**Figure 2 micromachines-13-01858-f002:**
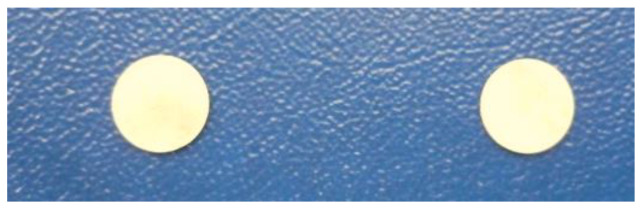
Current injection electrodes, made from Ag (silver).

**Figure 3 micromachines-13-01858-f003:**
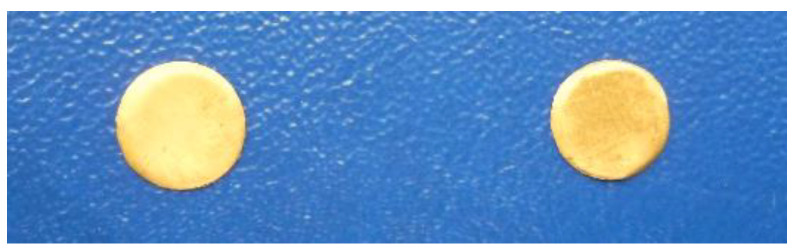
Electrodes of electric potential, made from Au (gold).

**Figure 4 micromachines-13-01858-f004:**
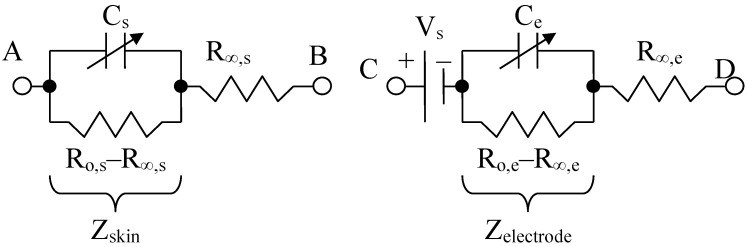
Skin or organ, Cole Model (left) and electrode Cole model (right), adapted from [47].

**Figure 5 micromachines-13-01858-f005:**
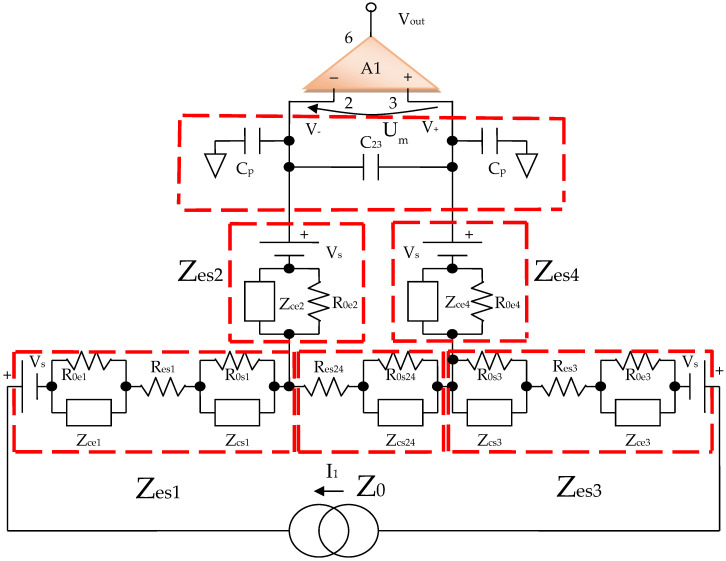
Electric diagram for modeling a system of four electrodes in contact with the skin and a target organ.

**Figure 6 micromachines-13-01858-f006:**
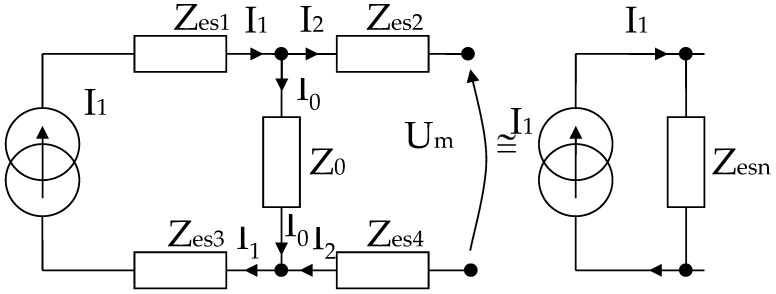
Simplified electric diagram for modeling a system of 4 electrodes in contact with the skin, placed at equal distances, and with identical shapes two by two.

**Figure 7 micromachines-13-01858-f007:**
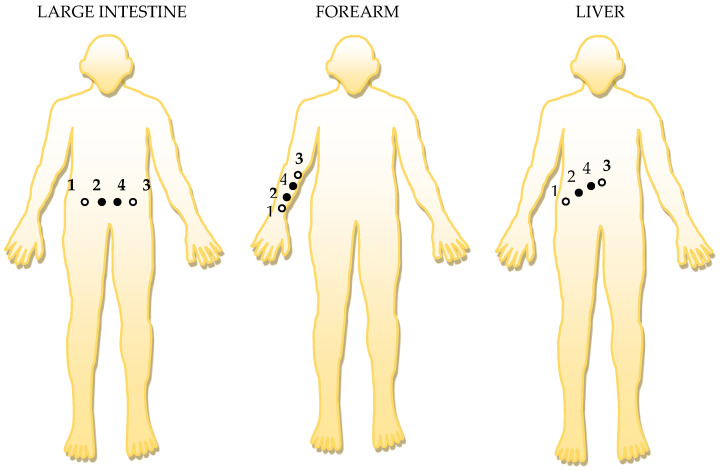
The positioning of the electrodes, in order, to targeted organ area: 1 and 3 are the current injection electrodes, made from Ag (silver); 2 and 4 are the electrodes of electric potential, made from Au (gold).

**Figure 8 micromachines-13-01858-f008:**
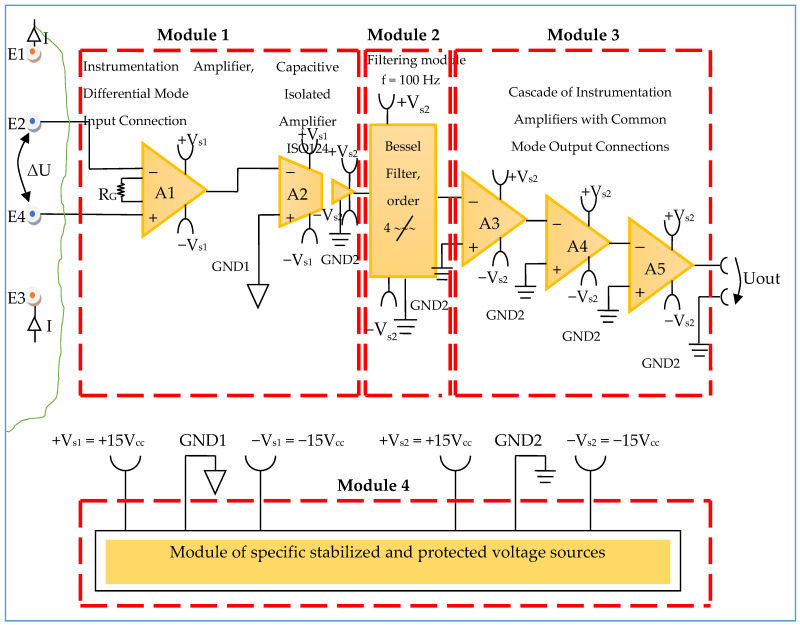
Block diagram of the electronic conditioning system used for acquiring biological signals.

**Figure 9 micromachines-13-01858-f009:**
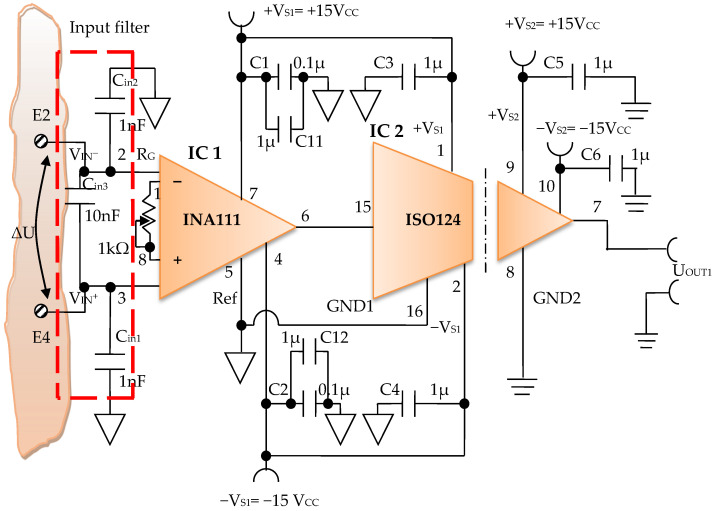
Electronic scheme of the electronic module for acquiring biological signals.

**Figure 10 micromachines-13-01858-f010:**
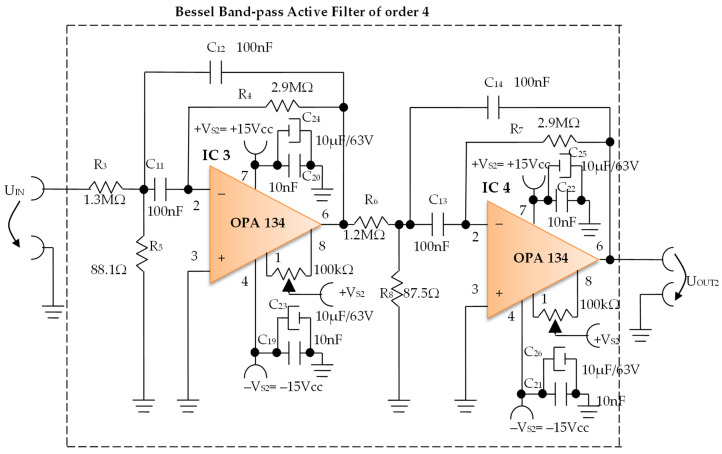
Electronic scheme of the Bessel band-pass filter module.

**Figure 11 micromachines-13-01858-f011:**
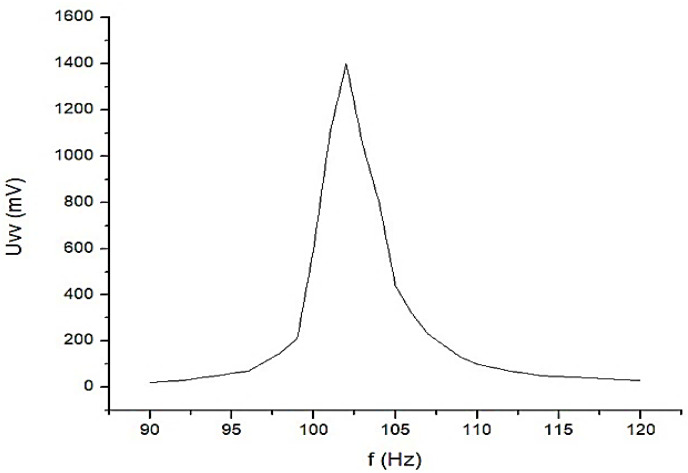
The graphical representation of the main characteristic of the 4th-order active Bessel band-pass filter.

**Figure 12 micromachines-13-01858-f012:**
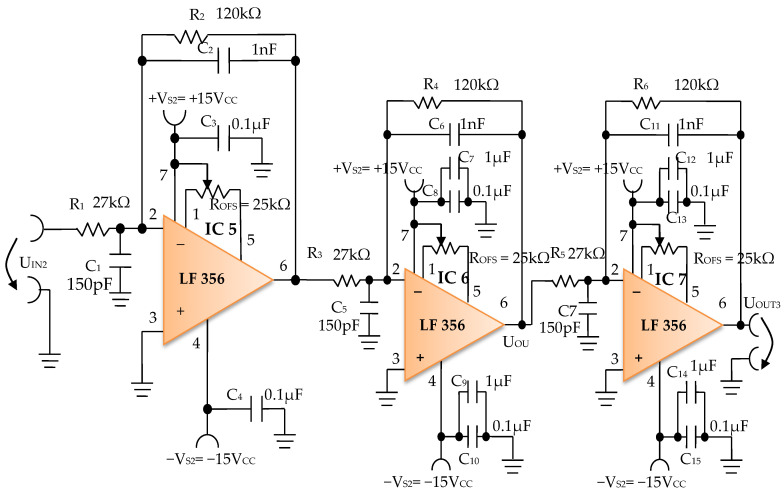
Electronic scheme of the common mode output amplifier electronic module.

**Figure 13 micromachines-13-01858-f013:**
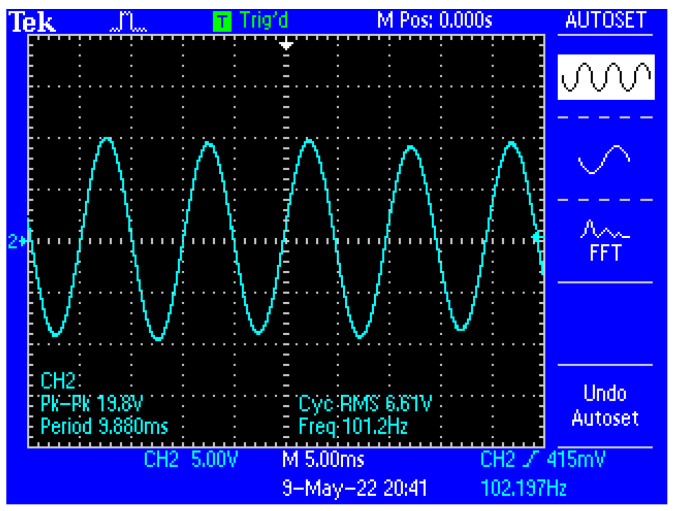
Capture of the waveform from the output of the electronic conditioning system, in the case of generating a sinusoidal signal with the characteristics: U_peak-to-peak_ = 10 mV and f = 100 Hz at the input.

**Figure 14 micromachines-13-01858-f014:**
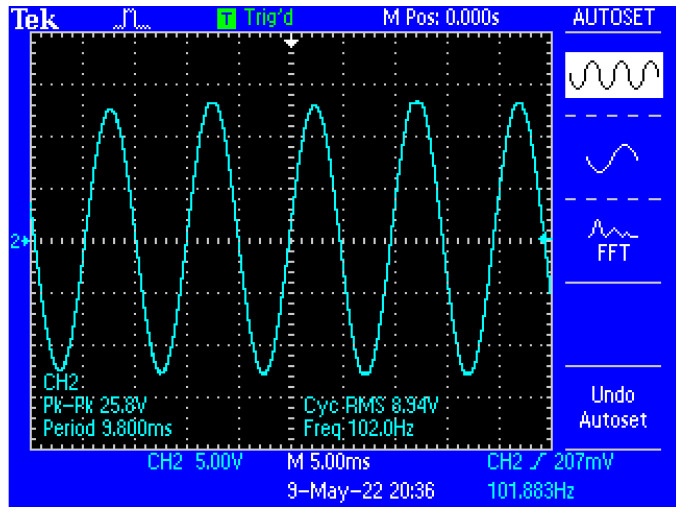
Capture of the waveform from the output of the electronic conditioning system, in case of generating a sinusoidal signal with the characteristics: U_peak-to-peak_ = 12.5 mV and f = 100 Hz at the input.

**Figure 15 micromachines-13-01858-f015:**
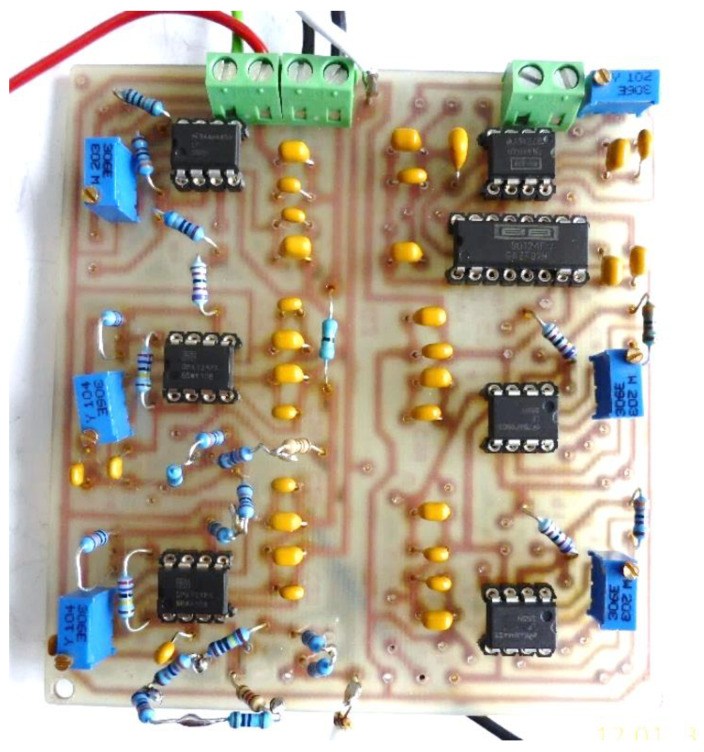
Electronic conditioning system used for acquiring and monitoring the biological signals, practical realization.

**Figure 16 micromachines-13-01858-f016:**
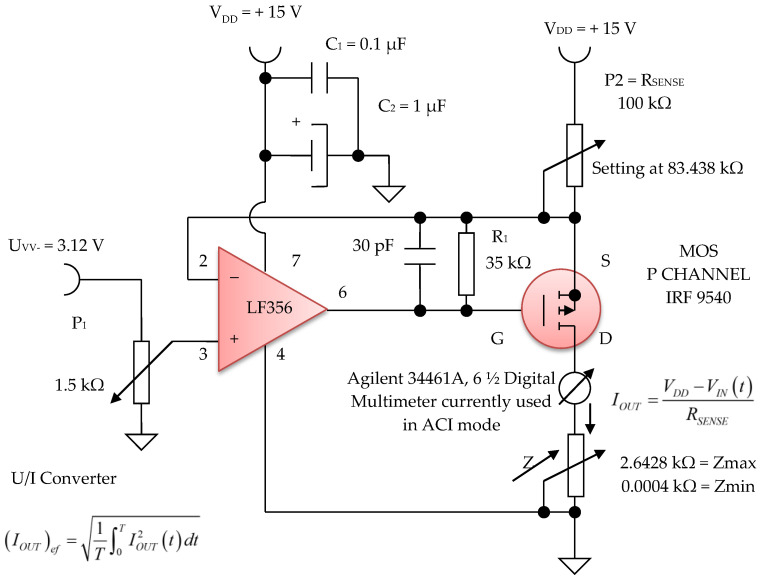
Voltage to constant current converter, 100 µA average value (average).

**Figure 17 micromachines-13-01858-f017:**
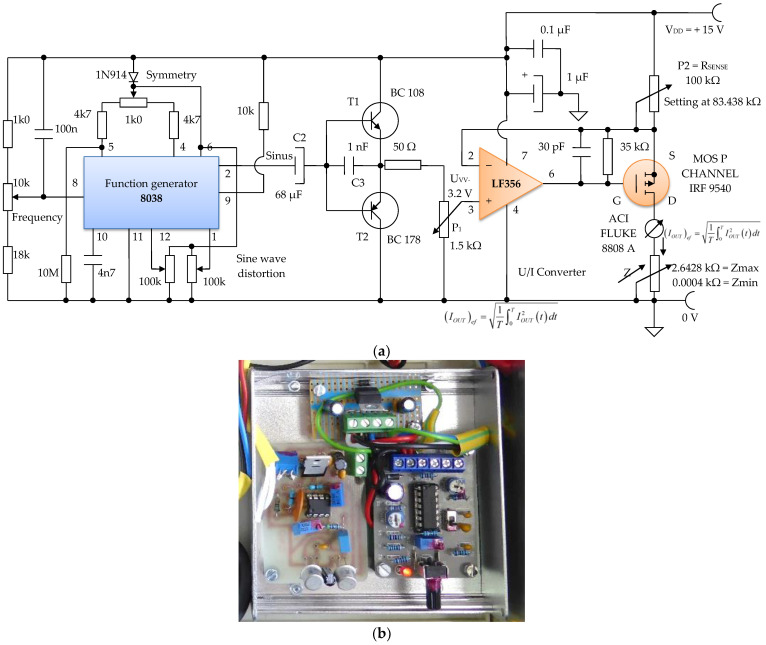
Electronic module of the constant current generator with an average value of *I*_1_ = 100 μA; (**a**) electronic scheme; (**b**) practical realization.

**Figure 18 micromachines-13-01858-f018:**
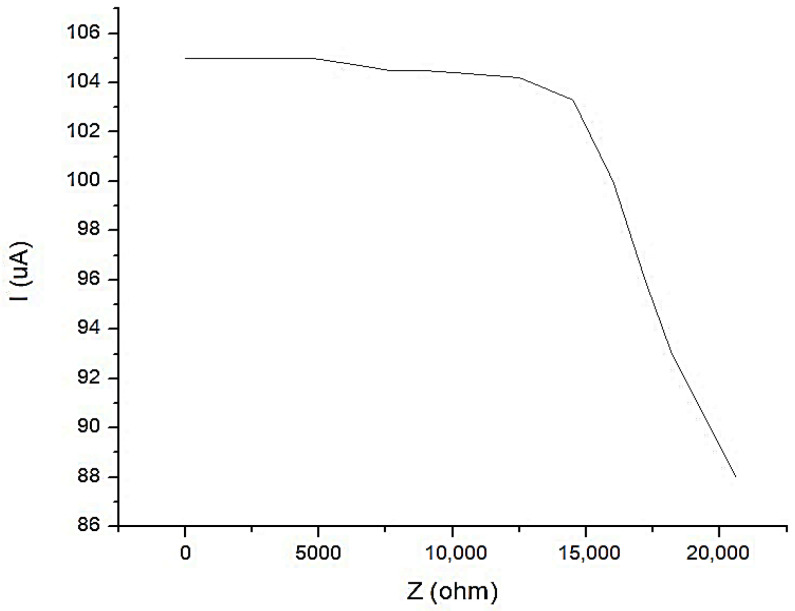
The characteristic of the constant current generator with an average value of *I*_1_ = 100 μA, when the current is depending on the load.

**Figure 19 micromachines-13-01858-f019:**
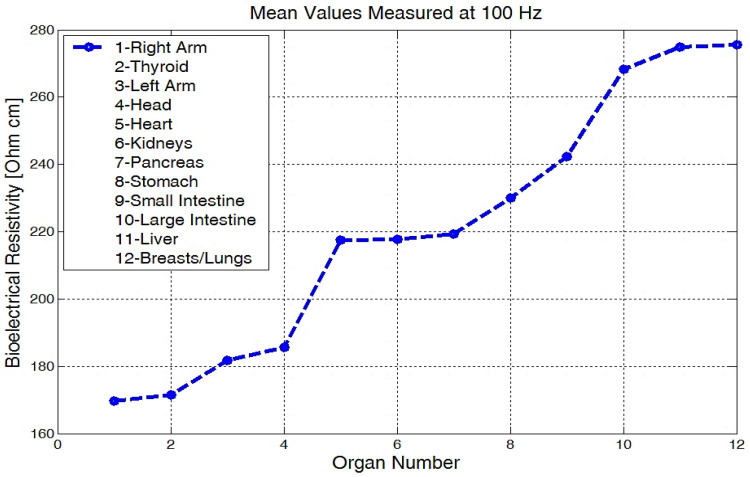
Mean values of bioelectrical resistivity measured at 100 Hz, for different body areas.

**Figure 20 micromachines-13-01858-f020:**
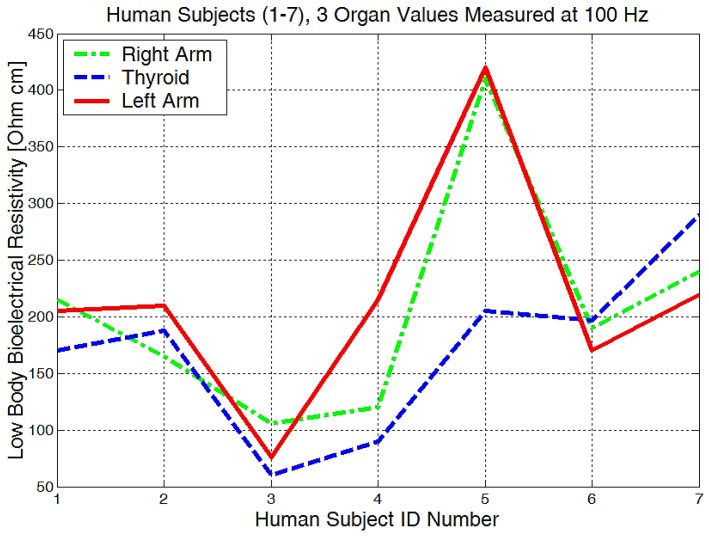
Lowest body resistivity zones represented for each human subject.

**Figure 21 micromachines-13-01858-f021:**
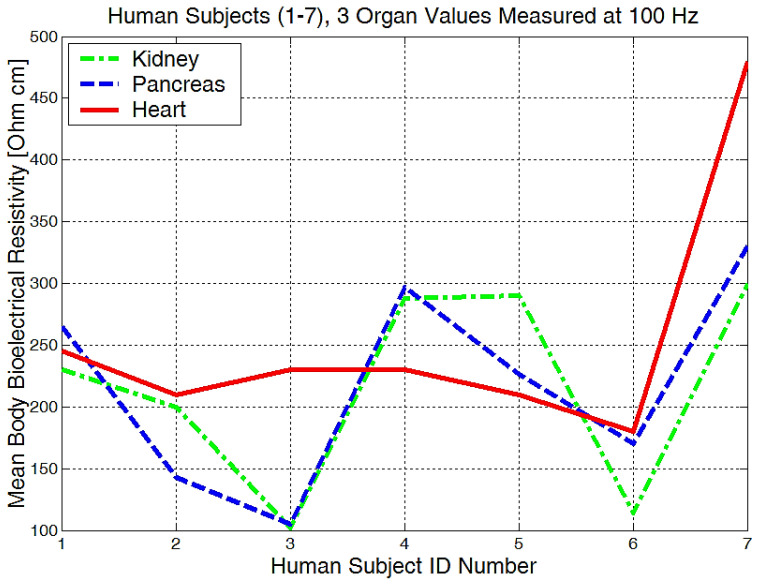
The mean body resistivity zones represented for each human subject.

**Figure 22 micromachines-13-01858-f022:**
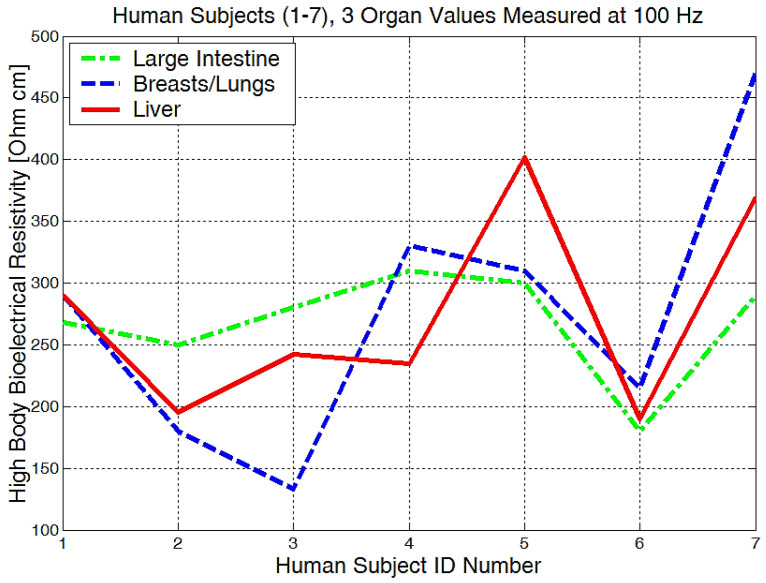
Highest body resistivity zones represented for each human subject.

**Figure 23 micromachines-13-01858-f023:**
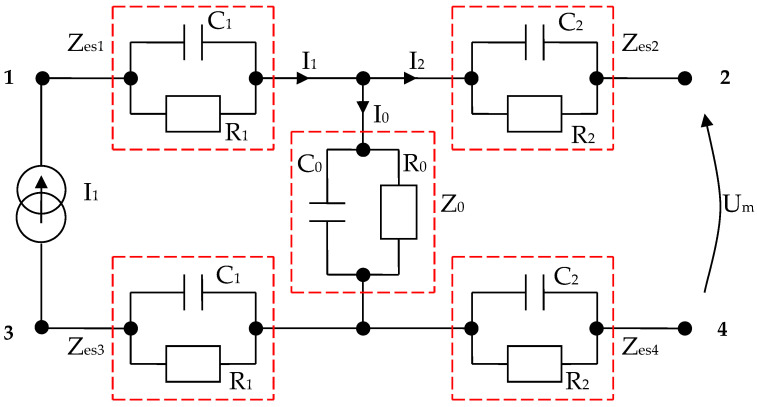
Electric diagram for modeling a system of four electrodes in contact with the skin and considering a target organ; each impedance is made from RC parallel disposed components.

**Figure 24 micromachines-13-01858-f024:**
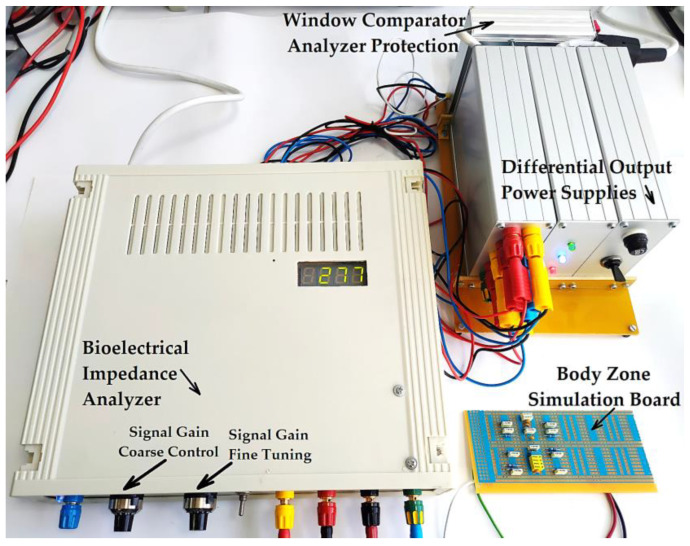
The practical achievement of the bioelectrical impedance analyzer, connected to the simulation circuit board that is calibrated for the liver organ.

**Table 1 micromachines-13-01858-t001:** RC series and parallel circuit measurements with the RLC bridge at 100 Hz, for 1 (Ag)-2 (Au) electrodes placed on the skin.

Skin–Electrode Parameters, Ag-Au (1-2)	R_s_ [kΩ]	C_s_ [pF]	Z_seq_ [kΩ]	R_0p_ [kΩ]	C_p_ [pF]	Z_peq_ [kΩ]	tanδ
Forearm	1120	2200	1333	2240	830	1377	0.57
Large Intestine	882	3380	999.4	2910	710	1395	0.67
Liver	665	3770	787.5	3650	690	1916	0.66

**Table 2 micromachines-13-01858-t002:** Series and parallel RC circuit measurements with RLC bridge at 100 Hz, for 3 (Ag)-4 (Au) electrodes placed on the skin.

Skin–Electrode Parameters, Ag-Au (3-4)	R_s_ [kΩ]	C_s_ [pF]	R_0p_ [kΩ]	C_p_ [pF]	tanδ
Forearm	1100	2250	2320	780	0.57
Large Intestine	920	3410	1660	1050	0.67
Liver	700	3880	1830	750	0.66

**Table 3 micromachines-13-01858-t003:** Series and parallel RC circuit measurements with RLC bridge at 100 Hz, for 2 (Au)-4 (Au) electrodes placed on the skin.

Skin-Sensors Parameters Au-Au (2-4)	R_s_ [kΩ]	C_s_ [pF]	R_0p_ [kΩ]	C_p_ [pF]	tanδ
Forearm	1590	980	2210	800	0.57
Large Intestine	1670	2830	2780	710	0.60
Liver	928	3280	3630	690	0.60

**Table 4 micromachines-13-01858-t004:** Series RC circuit measurements with RLC bridge from 40 kHz up to 300 kHz, for 2 (Au)-4 (Au) sensors placed on the skin.

Skin-Sensors Parameters R_es24_ [Ω] Au-Au (2-4)	f = 40 kHz	f = 100 kHz	f = 300 kHz
Forearm	1520	523	426
Large Intestine	1830	880	570
Liver	2610	930	620

**Table 5 micromachines-13-01858-t005:** Equivalent circuit measurements with the RLC bridge at 100 Hz, calculated values of impedance Zesn and measured values of bioelectrical resistivity with the bioelectrical impedance analyzer, summarized for four target body areas, namely forearm, kidney, large intestine, and liver.

Calibration and Accuracy of the Measurements	R_1_ [kΩ]	C_1_ [pF]	R_2_ [kΩ]	C_2_ [pF]	R_0_ [kΩ]	C_0_ [nF]	Z_0_ [kΩ]	Z_esn_ [kΩ]	ρ [Ωcm]
Forearm	6.172	981	6.179	985	11.95	221.24	6.164	16.456	180
Kidney	6.172	981	6.179	985	11.95	221.06	6.168	16.459	220
Large Intestine	6.172	981	6.179	985	11.95	220.74	6.174	16.461	265
Liver	6.172	981	6.179	985	11.95	220.65	6.176	16.463	277

## Data Availability

Not applicable.

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
