# Peer review of "Design and Manufacturing of Equipment for Investigation of Low Frequency Bioimpedance"

_micromachines, 2022, doi:10.3390/mi13111858_

Round 1
Reviewer 1 Report
The authors describe a technical solution for the low frequency measurement of electrical properties at the human body. Although the topic is worth to work on, there is nothing new in this manuscript.
The manuscript does not meet a minimal scientific standard.
Specific comments
The authors seem to have difficulties to distinguish between active signals and bioimpedance (several places in the manuscript).
The electrodes and the location of the electrodes are not defined. Are the current injecting and voltage monitoring electrodes mounted on a joint support (figure 3)?
Where were the electrodes located for assessment of the impedance of different organs?
It is impossible to prove that the authors got information about the inner organs because of the very limited number of test persons and the insignificance of the measurement due to the considerable influence of the skin. For instance, why is the impedance of the heart so high? It should be the lowest of all. But there is the lung in between! -
Why did the authors use Au and Ag-electrodes? Especially Ag-electrodes are questionable? Can they comment on the choice of the electrodes? Why did they not use Ag/AgCl- electrodes or polymerelectrodes (e.g. polypyrrol)?
Fig.1 is wrong. The connection to the differential amplifier should be directly at the voltage sensing electrodes.
The authors mention alpha- and gamma-dispersion without using it.
Eq.12. yields the magnitude of the impedance, not a total impedance. What is a total impedance?
In fig. 7, why do the authors use three amplifiers in module 3? Is it due to the lack of the right compounds?
Fig.8 Is there a good reason for using a WW2-compound like the INA111 ? Do the authors know about the MAX3000X – frontends? It works great for 100 Hz.
It is not clear whether the measurements with polymerelectrodes were made with the here described frontend or directly with a RLC-bridge.
Are the authors aware about current developments for current sources used in Bioimpedance measurements?
Fig.21 Why did the authors measure at 100 Hz and why did the not take the impedance spectrum for higher significance of the measurement?
Minor comments:
The English is not satisfactory, at least during the first pages.
The wording is confusing. Sometimes words are used which are common in BIA.
Reviewer 2 Report
Journal Name: Micromachines
Title: Design and Manufacturing of an Equipment for Investigation of Low Frequency Bioimpedance.
In the current article, the authors have made interesting research on low-frequency bioimpedance. The article will be interesting for Micromachine readers.
Article accepted in present form or with minor corrections.
- The author should trim the abstract
- Authors need to refer few earlier articles published in Micromachines and explain why the article is suitable for MDPI Micromachine Journal
- Authors need to further improve the literature review.
- The authors need to explain how it is possible to inject constant current this is quite clear.
- Is the authors take any special permission to perform these experiments since the research involves humans.
- Is there any possibility to use carbon electrode materials instead of metallic electrodes?
- What are the limitations of bioimpedance measurements?
- What are the error limits of the proposed methods/
- The conclusion can be further improved
- Please check the grammatical and syntax error
Reviewer 3 Report
Please, find the attached PDF file with my comments.

Round 2
Reviewer 1 Report
The manuscript shows some improvement but is far from being good. I will highlight a few places, where I guess, the authors are not familiar with bioimpedance measurements.
Abstract:
Give information on the frequencies used in the abstract
Line 57 A sensor does not develop a value but measures it. – please correct
Line 76 If a four electrode measurement is used, it is theoretically no difference, if the current electrodes are inside or outside. The same is true for the electrode material. However, in practical applications, we find the current injection electrode often as outer electrodes while the voltage sensing electrode are inside. For low frequency – applications, polarizable electrodes pose always problems. Therefore, non polarizable electrodes are preferred for measurements below 1 kHz. In the cited literature, polarizable electrodes are used above 10 kHz.
Line 80 What is a dry polarization electrode ? Do the authors mean a dry polarizable electrode ?
Line 82 PDMS is an insulator and not an electrode material. However, filled with graphite or CNT, it is a suitable and often used electrode material (PDMS-black).
Figure 1: Are the gray circles the electrodes ? If yes, the figure is wrong. Connect the A1 directly to the electrodes as you do it in the real setup.
Line 215 What is the value of the alpha-dispersion? Do the authors mean the circumference of the dispersion region? Please clarify!
Line 203-237 It is not clear, how you calculated the permittivity. For this, you should have a cell constant and the complex impedance. If you want to get an arc as you describe, you would need the spectrum. At what frequencies did you measure ? It seems, you measured only at 100 Hz? Please make very sure, what you really measured and -very important-, what frequencies you used when you used your new equipment.
Line 238 The relaxation time and time constant should be the same. Eq.9 comes from geometrical consideration of the permittivity while eq. 10 is the analytical approach what should be used. In general, the time constant can be only found using impedance spectroscopy and not at a single frequency.
Linen 720 – 727 The influence of the skin diminishes with frequency and is highest at low frequency.
Minor comments
Fill the author contributions
Fiig.13 and 14 Your sine wave looks horrible and much influenced by 50-Hz-noise.
Reviewer 3 Report
The authors have partially revised the original manuscript. A couple of issues, however, must still be addressed.
1 In the response to the Reviewers’ comments, the authors have said that a comparative analysis between the proposed circuit and a commercial impedance analyser is not possible because they have not such instrument in their laboratory. Nevertheless, without such measurements it is not possible to evaluate the accuracy of the measurements with the proposed device (i.e. the difference between the measured impedance and the real impedance). I think that the authors should at least make measurements with RC circuits realized with discrete components. That is: build RC circuits that model the measured bio-impedance; measure the values of the resistances and capacitances using laboratory instruments (i.e. multimeter, RLC bridge, etc.); calculate the impedance modulus at 100 Hz for the RC circuit; compare this value with the value measured by the proposed device.
2 A discussion must be added to explain how the proposed circuit improves the state of the art. Many on-board impedance measurement circuits have been proposed in literature that cover a wide range of frequencies, see for example “Grossi, M., Parolin, C., Vitali, B., & Riccò, B. (2019). Electrical Impedance Spectroscopy (EIS) characterization of saline solutions with a low-cost portable measurement system. Engineering Science and Technology, an International Journal, 22(1), 102-108” that covers the range 10Hz – 100 kHz. The authors should discuss how the proposed circuit improves the state of the art even if it makes measurements on a single frequency (100Hz).
Round 3
Reviewer 3 Report
The authors have revised the paper according to the Reviewer comments. I think the paper is now suitable for publication.